# Understanding the Impact of COVID-19 on Roma Vulnerable Communities in Western Romania: Insights and Predictive Factors from a Retrospective Study

**DOI:** 10.3390/v16030435

**Published:** 2024-03-12

**Authors:** Ionut Dragos Capraru, Catalin Marian, Dan Dumitru Vulcanescu, Sonia Tanasescu, Tiberiu Liviu Dragomir, Teodora Daniela Marti, Casiana Boru, Cecilia Roberta Avram, Monica Susan, Cristian Sebastian Vlad

**Affiliations:** 1Department of Epidemiology, “Victor Babes” University of Medicine and Pharmacy, Eftimie Murgu Sq. No. 2, 300041 Timisoara, Romania; ionut.capraru@umft.ro; 2Department of Biochemistry and Pharmacology, “Victor Babes” University of Medicine and Pharmacy, Eftimie Murgu Sq. No. 2, 300041 Timisoara, Romania; cmarian@umft.ro; 3Center for Complex Networks Science, “Victor Babes” University of Medicine and Pharmacy Timisoara, 300041 Timisoara, Romania; 4Department of Microbiology, “Victor Babes” University of Medicine and Pharmacy, Eftimie Murgu Sq. No. 2, 300041 Timisoara, Romania; dan.vulcanescu@umft.ro; 5Multidisciplinary Research Center on Antimicrobial Resistance (MULTI-REZ), Microbiology Department, “Victor Babes” University of Medicine and Pharmacy, Eftimie Murgu Sq. No. 2, 300041 Timisoara, Romania; 6Department of Pediatrics, “Victor Babes” University of Medicine and Pharmacy, Eftimie Murgu Sq. No. 2, 300041 Timisoara, Romania; tanasescu.sonia@umft.ro; 7Medical Semiology II Discipline, Internal Medicine Department, “Victor Babes” University of Medicine and Pharmacy, Eftimie Murgu Sq. No. 2, 300041 Timisoara, Romania; dragomir.tiberiu@umft.ro; 8Department of Medicine, “Vasile Goldis” University of Medicine and Pharmacy, 310414 Arad, Romania; boru.casiana@uvvg.ro; 9Department of Microbiology, Emergency County Hospital, 310037 Arad, Romania; 10Department of Residential Training and Post-University Courses, “Vasile Goldis” Western University, 310414 Arad, Romania; avram.cecilia@uvvg.ro; 11Centre for Preventive Medicine, Department of Internal Medicine, “Victor Babes” University of Medicine and Pharmacy, Eftimie Murgu Square, No. 2, 300041 Timisoara, Romania; susan.monica@umft.ro; 12Discipline of Pharmacology, Department of Pharmacology and Biochemistry, “Victor Babes” University of Medicine and Pharmacy, 300041 Timisoara, Romania; vlad.cristian@umft.ro

**Keywords:** COVID-19, SARS-CoV-2, Romania, Roma, Gypsy, laboratory, predictive values

## Abstract

Background: The COVID-19 pandemic disproportionately affected vulnerable populations like Roma patients in Western Romania due to marginalization and limited healthcare access. Methods: A retrospective study analyzed COVID-19 cases between March 2020 and August 2022 using data from the Directorate of Public Health in Timis county. Demographic, epidemiological, clinical, and laboratory data were assessed, along with risk factors and biomarkers for ICU admission and mortality prediction. The following biomarkers were assessed: C-reactive protein (CRP), ferritin (FER), IL-6, D-dimers, lactate dehydrogenase (LDH), high density lipoprotein cholesterol (HDL), and 25-OH vitamin D (25-OHD). Results: In comparison with the general population (GP), Roma patients were more overweight (*p* = 0.0292), came from rural areas (*p* = 0.0001), could not recall transmission source (*p* = 0.0215), were admitted to the intensive care unit (ICU, *p* = 0.0399) more frequently, had worse symptomatology (*p* = 0.0490), showed more elevated levels of CRP (*p* = 0.0245) and IL-6 (*p* < 0.0001) and lower levels of HDL (*p* = 0.0008) and 25-OHD (*p* = 0.0299). A stronger, significant correlation was observed between CRP and severity (rho = 0.791 vs. 0.433 in GP), and an inverse stronger significant one was observed between HDL and severity (rho = −0.850 vs. −0.734 in GP) in the Roma patients. The male sex continues to be an important risk factor for ICU admission (OR = 2.379) and death (OR = 1.975), while heavy smoking was more important in relation to ICU admission (OR = 1.768). Although the Roma ethnicity was 1.454 times more at risk of ICU admission than the GP, this did not prove statistically significant (*p* = 0.0751). CRP was the most important predictive factor in regards to admission to the ICU for both Roma (OR = 1.381) and the GP (OR = 1.110) and in regards to death (OR = 1.154 for Roma, OR = 1.104 for GP). A protective effect of normal values of HDL and 25-OHD was observed in the GP for both ICU admission (OR = 0.947, 0.853, respectively) and death (OR = 0.920, 0.921, respectively), while for the Roma group, normal 25-OHD values were only considered protective in regards to death (OR = 0.703). Cutoff values for ICU admission were 28.98 mg/L for Roma and 29.03 mg/L for GP patients, with high specificity for both groups (over 95). Conclusions: Higher rates of ICU admissions, severe symptomatology, and distinct laboratory biomarker profiles among Roma patients emphasize the critical importance of personalized care strategies and targeted interventions to mitigate the disproportionate burden of COVID-19 on vulnerable communities. CRP values at admission have had a clear impact as a risk assessment biomarker for Roma patients, while the significance of IL-6, HDL, and 25-OHD should also not be overlooked in these patients.

## 1. Introduction

Since its first appearance in Wuhan, China, in December 2019, COVID-19, caused by the severe acute respiratory syndrome coronavirus 2 (SARS-CoV-2), has rapidly spread across the globe, leading to widespread illness, death, and significant disruptions to society and economies. COVID-19 quickly evolved from a localized outbreak to a pandemic affecting almost every country [1].

The virus is primarily transmitted through respiratory droplets and has an incubation period of approximately 2–14 days. It can cause a wide range of symptoms, from mild respiratory illness to severe pneumonia, acute respiratory distress syndrome (ARDS), and death. Vulnerable populations, including the elderly and individuals with underlying health conditions, are at a higher risk of severe disease and mortality [2,3,4,5].

Excessive inflammation is a prominent characteristic of COVID-19, particularly in patients with severe disease. This inflammation is driven by an overactive immune response involving various cytokines. Several immune cells and inflammatory mediators have been identified in the illness process, including lymphokines, cytokines, monokines, tumor necrosis factors (TNF), and interferons [6,7]. The cytokine storm syndrome, which results from an overactive immune response to the virus, can lead to multiple organ dysfunction syndromes, disseminated intravascular coagulation, and the presence of venous thromboembolism and microthrombi in arterioles and venules in COVID-19 patient corpses [8,9,10,11].

It is important to note that every patient group is at risk of contracting SARS-CoV-2, and the cytokine storm and acute respiratory distress syndrome (ARDS) can develop in any COVID-19 patient, with the severity influenced by various known and unknown factors [12,13]. Other important biomarkers have been researched since the beginning of the pandemic, such as the C-reactive protein (CRP), interleukin 6 (IL-6), hepatic, pulmonary, cardiac, renal, and iron markers. Of all researched potential biomarkers, some, including CRP, IL-6, D-dimers lactate dehydrogenase (LDH), 25-OH vitamin D (25-OHD), ferritin (FER), and high-density lipoprotein cholesterol (HDL), have proven a better capacity to properly assess severity, risk of ICU admission, and even death [12,13,14,15,16].

The Roma group, also known as Romani, Gypsy or Tigani, are one of Europe’s most significant ethnic minority groups, estimated to be over 10 million people. The Roma population is widely dispersed across Europe, with significant communities in countries such as Romania, Bulgaria, Hungary, Slovakia, and Spain [17]. Western Romania, in particular, hosts a sizable Roma population, often residing in marginalized and segregated settlements on the outskirts of towns and cities. These settlements, commonly referred to as “Roma camps” or “Roma villages”, tend to lack basic infrastructure and access to essential services, perpetuating the cycle of poverty. Persistent discrimination in education, employment, and housing has resulted in high levels of poverty and social exclusion. Limited access to quality education and vocational training hinders their economic mobility, trapping many Roma individuals and families in a cycle of intergenerational poverty [18,19].

The healthcare challenges facing the Roma in Western Romania are intimately tied to their socio-economic circumstances. Roma communities experience disparities in access to healthcare services, resulting in poorer health outcomes [20]. Discrimination and cultural barriers can deter Roma individuals from seeking medical care, leading to delayed diagnoses and treatment. Moreover, the prevalence of chronic diseases, such as diabetes, hypertension, and respiratory illnesses, is higher among the Roma population due to factors like limited access to healthcare, poor nutrition, and overcrowded living conditions. Maternal and child health outcomes are also adversely affected, with higher rates of infant mortality and lower rates of prenatal care utilization compared to the general population [21]. This limited access to proper education and healthcare may be the factor at play in regards to the presumed hesitation of this population towards vaccination [22,23].

As such, the main objective of the present study is to analyze the impact of COVID-19 on Roma patients from Western Romania, hospitalized in Timisoara, their demographic, epidemiological, clinical and laboratory data, and risk factors, according to the data provided by the regional Directorate of Public Health of the Timis county. Secondly, biomarkers were selected, as further described, in order to assess their role as potential tools for assessing the risk of ICU admission and death, and to seek differences between the Roma group and the general population of Romania.

## 2. Materials and Methods

### 2.1. Study Design

The current study follows an observational retrospective register-based design and features patients hospitalized in Timisoara between March 2020 and August 2022. The investigation is based upon the database of the Directorate of Public Health of the Timis county, as it contains all reported cases in regards to COVID-19 diagnosed in this county, both at home and in hospitals. The objective of this study was to retrospectively investigate COVID-19 cases by gathering data from available medical records of patients who were positive during the study period. Vaccination status was verified using the QR code certificate issued in Romania, in accordance with the COVID-19 pandemic regulations established by the European Union.

The research adhered to the ethical standards set by the Directorate of Public Health of the Timis county and received approval from the ethics committee under the approval number: 27994/14.NOV.2023. The patient’s signed agreement was not necessary as the study is retrospective in nature and the data were gathered from the county registry. 

### 2.2. Inclusion Criteria and Selected Data

Patients were included if they matched the following criteria: (1) records mentioning ethnicity; (2) positive result at the polymerase chain reaction test (PCR) from oropharyngeal and nasal swabs; (3) a complete data set, as presented below. No personal data were recorded. The clinical severity of SARS-CoV-2 infection was categorized into several levels based on clinical characteristics and laboratory findings. Exclusion was based on lack of data, therefore, patients that were positive for COVID-19 but were not admitted to the hospital are not present.

These categories included the following: (1) Asymptomatic: cases with a positive RT-PCR test but no clinical symptoms; (2) Mild: cases with upper respiratory tract infection symptoms like fever, fatigue, myalgia, cough, and sore throat; (3) Moderate: patients with pneumonia who had fever and a cough but did not exhibit dyspnea or hypoxemia symptoms; (4) Severe: individuals who initially had fever and a cough but later developed dyspnea and central cyanosis within a week, characterized by arterial oxygen saturation below 92%, OR presented with more than 50% of the lungs exhibiting ground-glass opacities on chest X-ray or CT scans; (5) Critical: cases that rapidly progressed to acute respiratory distress or respiratory failure, and were at risk of developing complications such as shock, encephalopathy, myocardial issues, coagulation dysfunction, and acute kidney injury.

The following demographic data were obtained. Age, sex, location, body mass index (BMI), and cigarette and alcohol intake. Regarding COVID-19 information, vaccination status and transmission source were also noted. Investigated laboratory data were as follows: C-reactive protein (CRP), ferritin (FER), IL-6, D-dimers, lactate dehydrogenase (LDH), high density lipoprotein cholesterol (HDL), and 25-OH vitamin D (25-OHD). 

Normal values were as follows: CRP (<5 mg/L); FER (0–1 year: 12–327 ng/mL, 1–7 years: 4–67 ng/mL, F: 7–17 years: 7–84 ng/mL, >17 years: 13–150 ng/mL, M: 7–17 years: 14–152 ng/mL, >17 years: 30–400 ng/mL); IL-6 (<7 pg/mL); D-dimers (<250 ng/mL); LDH (0–1 year: 225–600 U/L, 1–17 years: 120–300 U/L, >17 years: 135–225 U/L); HDL (>40 mg/dL); 25-OHD (>30 ng/mL).

### 2.3. Statistical Analysis

Minimum sample calculation was performed using the G*Power software (v 3.1.9.6), using an a priori test to calculate the minimum sample size for a small effect size (0.3) and a power of 95%. The resulting necessary sample was 506 patients. As such, our sample of 578 patients was considered satisfactory.

The MedCalc Statistical Software, version 20.218 (MedCalc Software bv, Ostend, Belgium) was used to conduct the statistical analysis. The Shapiro–Wilk test was used to evaluate all continuous variables for normal distribution. As most data were not normally distributed, variables were expressed as median and interquartile range (IQR) and were compared using the Mann–Whitney test.

Contingency tables were prepared and then analyzed using the Chi^2^ test to look for correlations between the presence of clinical severity and the status of the investigated biomarkers. Odds ratio was also assessed using contingency tables.

To check for correlations in regards to ethnicity, the Spearman’s rank correlation was used. The following interpretation was considered: r ≤ 0.10 was considered “very weak”, 0.10 < r ≤ 0.33 was considered “weak”, 0.33 < r ≤ 0.66 was considered “moderate”, and r > 0.66 was considered “strong”. A multivariate logistic regression analysis was performed to determine the influence of patients’ ethnicity on outcome and for parameters at hospital admission. 

Any of the tested parameters’ diagnostic value in predicting ICU admission and the probability of COVID-19-related death was evaluated using the receiver operating characteristic (ROC) curve. The Youden index was used to establish the cutoff point and the area under curve (AUC) is also provided. The DeLong et al. methodology was selected. Additionally, a comparison of separate ROC curves was conducted. Statistical significance was set at *p* values of 0.05 or lower for all tests.

## 3. Results

A total of 578 patients met the inclusion criteria, presenting a complete set of the selected data. There were 144 (24.91%) Roma patients and 434 (75.09%) Romanian patients. Generally, there were 243 (42%) females and 335 (58%) males, and the median age was 57.52 (IQR = 31.07). Demographic data are presented in Table 1. Significant differences between the studied groups were in regards to BMI (*p* = 0.0292), being higher for the Roma group, location (*p* = 0.0001), with more Roma patients coming from rural living conditions, and transmission source (*p* = 0.0215), with more unknown epidemiological links in the Roma population. 

Rho and *p* values from the Spearman test in regards to the Roma ethnicity are also provided. Statistically significant correlations resulting from this test were observed in regards to BMI (positive weak correlation) and rural place of origin (positive weak correlation).

For the whole lot, there were 95 (16.4%) asymptomatic cases, 145 (25.1%) mild, 148 (25.6) moderate, 178 (30.8%) severe, and 12 (2.1%) critical (Table 2). Regarding this aspect, a significant difference was observed between Roma and non-Roma populations, with Roma patients presenting with more severe symptomatology (*p* = 0.0490). Overall, there were 200 (34.6%) patients admitted to the ICU, of which the percentage of Roma proved higher (*p* = 0.399). Regarding mortality, there were 114 (19.7%) total cases and no differences were observed between the studied groups (*p* = 0.1764). Spearman test values are provided, as well. A statistically significant correlation was observed between Roma ethnicity and ICU admission (very weak). Another very weak correlation with severity did not achieve statistical significance, although the *p* value was very small (*p* = 0.0506). 

Th odds ratio was 1.500 (95% CI: 1.0180–2.2103) for the Roma group, suggesting a statistically significant elevated risk for ICU admission in this group (*p* = 0.0404), compared to the general population of Romanians. Regarding death, the odds ratio was similar, at 1.3677 (95% CI: 0.8680–2.1552); however significance level was not conclusive (*p* = 0.1771).

These parameters were available for all selected patients and are among the most consistent in regards to severity in COVID-19, as will be discussed later. Notable differences were observed between CRP (*p* = 0.0245), IL-6 (*p* < 0.0001), HDL (*p* = 0.0008) and 25-hidroxy-vitamin D (*p* = 0.0299), with the first two being more elevated in the Roma group, while the other two were lower (Table 3). The results of the Spearman test indicated a few statistically significant correlations with the Roma ethnicity. CRP values proved a very weak relationship, IL-6 values proved a weak relationship, HDL proved an inverse weak relationship, and 25-OHD proved an inverse very weak relationship. 

Afterwards, to check for possible links between these analytes and severity, a correlational analysis was performed for both groups (Table 4). All links were found to be statistically significant in regards to severity. The following were considered direct and moderate: D-dimers for Roma and CRP, FER, and D-dimers for the general population. The following were considered direct and strong: CRP, IL-6, LDH for Roma and IL-6, and LDH for the general population. All inverse relationships were strong. It is important to note that there is a big difference between the rho values for CRP and HDL between the two groups, while the rest were similar.

Multivariate logistic regression analysis was performed to determine the influence of patients’ ethnicity on ICU admission and death, which were considered dependent variables, while the other categorical variables were considered independent variables (Table 5). This confirms the male sex as a predictor for both ICU admission and death, being around 2 times more likely than females, while old age was presented a β coefficient of 2.696 in regards to death, when compared to adults. Another important result was that of the smoking status, which showed a statistically significant increased chance of ICU admission for heavy smokers (β = 1.768). For this analysis, although the β was around 1.4 for each outcome, the influence did not reach statistical significance in regards to ethnicity. 

Considering that a high BMI has been previously associated with severity, association testing was also carried out (Table 6). The results show a clear association between severity and weight distribution in the Roma group (Chi^2^ *p* = 0.0347), with the relationship being considered weak and direct. However, no statistically significant associations could be made in the general group. 

Spearman’s rho was once again used to assess the relationship between severity and ICU admission and death, respectively, for the whole lot and each separate group (Table 7). All correlations proved significant, direct, and moderate. Correlation analysis between outcomes also suggests a similar death rate between the two populations (Table 8). The correlation between relationships proved to be moderate and direct.

Afterwards, the multivariate analysis was performed to check the relationship between ethnicity and elevated inflammatory markers at hospital admission, in regards to ICU admission (Table 9) and death (Table 10).

The following test was the ROC curve analysis for each parameter, in order to attempt obtaining an optimal threshold for the risk of further ICU admittance and even death (Table 11). Notably, the highest AUC was for CRP in the Roma group in regards to ICU admission (AUC = 0.985, Se = 93.33, Sp = 96.43, cutoff values: 28.98). This parameter proved a difference between the two groups’ AUC and although the cutoff values are similar (Roma = 28.98 vs. Romanian = 29.03), due to the elevated AUC and sensitivity, the Roma group value can be considered more accurate. No other differences were observed. All individual ROC tests were statistically significant (AUC different than 0.5) and they can be observed graphically in the Appendix A.

## 4. Discussion

In addition to exposing the flaws in our international health systems, the COVID-19 pandemic has brought to light the already-existing inequalities across different social groups. During this crisis, vulnerable populations—defined by variables including socio-economic position, race, and access to healthcare—have been disproportionately affected [2,3,4]. Research consistently highlights the heightened vulnerability of certain populations to severe outcomes from COVID-19. Studies such as those conducted by Khanijahani et al., Lewis et al., and Chilunga et al. emphasize the increased risk faced by individuals with pre-existing health conditions, which is often prevalent in vulnerable communities [24,25,26]. 

Limited access to healthcare services exacerbates this risk, emphasizing the urgent need for targeted interventions to ensure equitable health outcomes. Of course, these populations are also at a disadvantage from economic, social, educational and psychological hardships and inequalities, effects visible in both the international communities and the population of Romania [27,28,29,30].

Roma and Gypsy populations have historically encountered systemic discrimination, economic marginalization, and limited access to healthcare, placing them in a particularly vulnerable position during the ongoing pandemic. The challenges faced by these communities intersect with the broader socio-economic factors that contribute to health disparities, exacerbating the impact of COVID-19 [20,22,31]. Some communities and media outlets have even adopted an exaggerated narrative of how Roma patients are “bearers of COVID” [32,33]. There are several studies, even at an international level, in regards to the social aspect of the Roma communities during the pandemic; however, there are only a handful that have carried out studies from a medical point of view on this category of ethnic patients. Their situation has been observed in the European Union, as well [34,35,36].

Generally, the health results for Roma populations are significantly poorer; they include significantly lower life expectancies, a higher prevalence of mental and physical health issues, and a higher uptake of dangerous health behaviors. Due to their tendency to live in close quarters with several family members, Roma communities are disproportionately affected by infectious illnesses including measles, hepatitis, and tuberculosis [37].

Regarding demographic modifications, a difference in BMI, location of residence, and transmission source was observed. Roma patients tended to be more overweight (53.47% vs. 40.78% in the general population), come from a rural place (60.42% vs. 41.71% in the general population) and could not state the origin of transmission more frequently (50% vs. 44.01%). Similar findings were also observed in the studies of Mocanu et al., while also taking into consideration that Roma patients were more likely to be unemployed [38,39]. The unemployment situation of Roma patients remains a hot topic, even at an international level [40]. Their study, published in 2023, also established more Roma patients being chronic smokers, while this study could not establish such a relationship (*p* = 0.1546).

There were only 5 (3.47%) minors in our Roma group and 21 (4.84%) in our general population group. These patients were excluded in the research by Mocanu et al. [38,39]; however, there are two studies by Miconi et al. discussing the demographics, risks, and perspectives of Roma adolescents [41,42].

On the subject of vaccination, a difference was not established between the two groups, which was also observed in the work of Mocanu et al., although they did notice a higher use of the Jcovden (Ad26.COV2.S) vaccine in the Roma group and a preference for the Pfizer (BNT162b2) vaccine in the Romanian group. While the Moderna (mRNA-1273) vaccine use was similar, this was not statistically significant (*p* = 0.757) [39]. Although a general vision of vaccine hesitancy was observed [43], usually due to the mobility of the group (emigrant/traveler status), their relative hard to reach locations, poor access to healthcare services, stereotypes, etc. [44,45], there are articles that also claim that a decent amount of Roma were interested in getting vaccinated and did so, as long as access was provided to their communities directly [46,47]. 

In the present study, there were more patients with more severe (37.50% vs. 28.57% in the Romanian group) or critical symptomatology (4.17% vs. 1.38% in the general population group), with a *p* value of 0.0490. This is consistent with the findings from Mocanu et al.’s research, which has observed a longer hospital stay [38,39], longer viral clearance, and more severe imaging features [39] in this group. Also, ICU admission was more significant in the Roma group; however, mortality was not statistically significant, which is in line with the findings from Mocanu et al. [38,39]. Based on the odds ratio, the risk for a Roma of getting admitted to the ICU was 1.5 times than the general Romanian population. 

The demographic characteristics of the lot admitted to the ICU (Roma = 60, Romanian = 140) were similar, with similar results at the Chi^2^ tests, with the exception of severity, which could not achieve statistical significance (*p* = 0.1510) even though there were more severe (71.67% vs. 65.71% in the Romanian group) and critical patients (8.33% vs. 2.86% in the Romanian group).

The analysis of laboratory results provides valuable insights into the severity of the infection, helps guide treatment decisions, and monitors the overall health status of individuals. Several biomarkers that have consistently been used to assess severity, ICU admission, and death are as follows: CRP, IL-6, D-dimers, LDH, 25-OHD, ferritin, and HDL cholesterol. Generally speaking, for CRP, IL-6, D-dimers, LDH, and ferritin, the more severe cases have had more elevated values, while HDL cholesterol and 25-OHD have proven an inverse relationship to severity [12,13,14,15,16].

While all parameters were out of the normal range, a comparison of median values revealed the following statistically significant modifications in the Roma group: elevated CRP and IL-6, decreased HDL and 25-OHD, when compared to the general population group. The elevated values can also be observed in both studies by Mocanu et al. [38,39]. However, the recorded HDL values did not differ in their earlier study [38], while being absent in the later one. 25-OHD was not studied by the research group of Mocanu et al.; however, previous research has noted its impact in COVID-19 [15,16]. While still on the subject of vitamin D, it is important to note that the median level for the whole lot was 20.82, which is considered the threshold value between normal values and deficiency, with many authors claiming that the optimal levels should be between 30 and 50 ng/mL [4,14,16]. As such, considering that half of the patients were under this value, supplementation of vitamin D is recommended. 

Firstly, these values were tested using the ROC/AUC tests to check their role as prediction tools for both ICU admission and death, based on parameters obtained at hospital admission. As such, all studied parameters in the hospital admission lot had an AUC different than 0.5, which translates to their good use as predictors, for both ICU admission and death, with similar findings from a previous study focusing on their utility [16]. When comparing the AUCs between Roma and non-Roma, only the CRP proved statistically significant in regards to ICU admission, although the cutoff point was similar (28.98 vs. 29.03, respectively). However, this is due to a higher sensitivity in the Roma group (93.33 vs. 65.7 in the general population lot).

### Strengths and Limitations

Regarding the strengths of the study, several can be discussed. The inclusion of a specific ethnic minority group, such as the Roma population, adds diversity to the study and allows for the examination of health disparities and vulnerabilities in marginalized communities. Incorporating laboratory data and biomarkers for ICU admission and mortality prediction enhances the depth of the analysis and provides valuable insights into the physiological responses and risk factors associated with COVID-19.

The study’s focus on identifying risk factors, biomarkers, and predictors of severe COVID-19 outcomes among the Roma population has direct clinical relevance and can inform personalized care strategies and interventions for this vulnerable group.

Statistically, the combined use of logistic regression and ROC curve analysis helps in identifying and determining the sensitivity, specificity, and optimal threshold for the studied analytes in the selected groups.

On the other hand, as with all studies, potential limitations have also been identified. As this paper is based on data from the county registry system, to which all hospitals within Timis county had reported their data, one important limitation to discuss is the fact that laboratory equipment differs from one hospital to another. As such, it is important to take into account that each hospital may report different results based on the specific equipment and reagents they use.

Another limitation might be the nature of retrospective studies, which might overlook cases with limited data. Comorbidities and treatment were not available nor assessed and may be regarded as confounding factors. Another aspect that this article has not taken into account is the temporal variation (number of waves) and testing for different variants of SARS-CoV-2, which was limited in our country.

Lastly, the study population of 144 Roma patients may be considered relatively small for drawing definitive conclusions, especially when further divided into smaller groups for statistical evaluation. The subgroup analyses within the Roma population may have limited statistical power to detect significant differences or associations due to the smaller sample sizes.

## 5. Conclusions

The consequences of COVID-19 on communities that are already at risk go well beyond the immediate health crisis. A comprehensive and focused strategy that takes into account the social, educational, economic, and health aspects is needed to address the inequities. By integrating laboratory data into the clinical decision-making process, healthcare professionals can better address the unique health needs of Roma patients, contributing to more personalized and effective care strategies during the COVID-19 pandemic. 

In this study, Roma patients were overweight, came from rural areas, could not recall the transmission source, were admitted to ICU more frequently, showed worse symptomatology, and showed more elevated levels of CRP and IL-6 and lower levels of HDL cholesterol and 25-hidroxy-vitamin D. The studied laboratory parameters at hospital admission had predictive values in regards to ICU admission and mortality, with CRP being more sensitive for Roma patients. Further research and data collection are necessary to address the limitations of the study and develop more effective interventions.

## Figures and Tables

**Table 1 viruses-16-00435-t001:** Demographic data and results of the Chi^2^ tests for the whole lot.

Variable	Roma (n = 144)	Romanian (n = 434)	*p* Value
**Sex**			**0.9286**
Male	83 (57.64%)	252 (58.06%)	
Female	61 (42.36%)	182 (41.94%)	
Spearman rho (male)	−0.0037		0.9287
**Age**			**0.5753**
<18	5 (3.47%)	21 (4.84%)	
18–40	30 (20.83%)	70 (16.13%)	
40–65	61 (42.36%)	192 (44.24%)	
>65	48 (33.33%)	151 (34.79%)	
Spearman rho	−0.0150		0.7190
**BMI**			**0.0292 ***
Underweight	8 (5.56%)	30 (6.91%)	
Normal	59 (40.97%)	227 (52.30%)	
Overweight	77 (53.47%)	177 (40.78%)	
Spearman rho	0.1060		0.0107 *
**Location**			**0.0001 ***
Urban	57 (39.58%)	253 (58.29%)	
Rural	87 (60.42%)	181 (41.71%)	
Spearman rho (rural)	0.1620		0.0001 *
**Tobacco intake**			**0.1546**
No/occasional smoking	42 (29.17%)	140 (32.26%)	
Regular smoker	50 (34.72%)	174 (40.09%)	
Heavy smoker	52 (36.11%)	120 (27.65%)	
Spearman rho	0.0635		0.1274
**Alcohol intake**			**0.5215**
No/occasional drinking	69 (47.92%)	201 (46.31%)	
Regular drinker	65 (45.14%)	189 (43.55%)	
Heavy drinker	10 (6.94%)	44 (10.14%)	
Spearman rho	−0.0627		0.2730
**Transmission source**			**0.0215 ***
Community	13 (9.03%)	82 (18.89%)	
Family	59 (40.97%)	161 (37.10%)	
Unknown	72 (50.00%)	191 (44.01%)	
Spearman rho (unknown)	0.0520		0.2116
**Vaccination**			**0.5549**
Yes	19 (13.19%)	66 (15.21%)	
No	125 (86.81%)	368 (84.79%)	
Spearman rho	−0.0246		0.5553

*: *p* < 0.05, statistically significant.

**Table 2 viruses-16-00435-t002:** Severity and presentation data and results of the Chi^2^ tests.

Variable	Roma (n = 144)	Romanian (n = 434)	*p* Value
**Severity**			**0.0490 ***
Asymptomatic	23 (15.97%)	72 (16.59%)	
Mild	31 (21.53%)	114 (26.27%)	
Moderate	30 (20.83%)	118 (27.19%)	
Severe	54 (37.50%)	124 (28.57%)	
Critical	6 (4.17%)	6 (1.38%)	
Spearman rho	0.0814		0.0506
**ICU admission**			**0.0399 ***
Yes	60 (41.67%)	140 (32.26%)	
No	84 (58.33%)	294 (67.74%)	
Spearman rho	0.0855		0.0398 *
**Deceased**			**0.1764**
Yes	34 (23.61%)	80 (18.43%)	
No	110 (76.39%)	354 (81.57%)	
Spearman rho	0.0563		0.1766

*: *p* < 0.05, statistically significant.

**Table 3 viruses-16-00435-t003:** Comparison of laboratory data at hospital admission.

Variable		Roma (n = 144)	Romanian (n = 434)	*p* Value
CRP	Median	25.245	20.945	**0.0245 ***
	IQR	24.760	18.080	
Spearman rho		0.0936		0.0244 *
FER	Median	184.225	173.770	**0.9548**
	IQR	219.315	163.820	
Spearman rho		0.0024		0.9548
IL-6	Median	8.495	7.105	**<0.0001 ***
	IQR	5.040	3.360	
Spearman rho		0.2400		<0.0001 *
D-dimers	Median	274.895	261.000	**0.3228**
	IQR	154.185	195.910	
Spearman rho		0.0412		0.3232
LDH	Median	211.180	202.800	**0.8921**
	IQR	98.835	127.920	
Spearman rho		0.0057		0.8923
HDL	Median	36.160	39.695	**0.0008 ***
	IQR	14.060	12.310	
Spearman rho		−0.1400		0.0008 *
25-OHD	Median	20.265	20.995	**0.0299 ***
	IQR	8.700	10.410	
Spearman rho		−0.0904		0.0298 *

CRP = C-reactive protein, FER = ferritin, IL-6 = interleukin-6, LDH = lactate dehydrogenase, HDL = high density lipoprotein cholesterol, 25-OHD = 25-hidroxy-vitamin D, *: *p* < 0.05, statistically significant.

**Table 4 viruses-16-00435-t004:** Correlation analysis between the groups for the selected biomarkers.

	Roma		Romanian	
	rho	*p*	rho	*p*
CRP	0.791	<0.0001 *	0.433	<0.0001 *
FER	0.661	<0.0001 *	0.633	<0.0001 *
IL-6	0.802	<0.0001 *	0.897	<0.0001 *
D-dimers	0.647	<0.0001 *	0.610	<0.0001 *
LDH	0.811	<0.0001 *	0.861	<0.0001 *
HDL	−0.850	<0.0001 *	−0.734	<0.0001 *
25-OHD	−0.862	<0.0001 *	−0.868	<0.0001 *

*: *p* < 0.05, statistically significant.

**Table 5 viruses-16-00435-t005:** Results of logistic regression analysis for demographic variables.

	ICU EXP (β)	95% CI Low	95% CI High	*p*	Death EXP (β)	95% CI Low	95% CI High	*p*
Female	1	-	-	-	1	-	-	-
Male	2.379	1.598	3.541	<0.0001 *	1.975	1.220	3.199	0.0057 *
Adults (18–64)	1	-	-	-	1	-	-	-
Children (<18)	0.686	0.215	2.191	0.5242	1.002	-	-	0.9980
Elders (≥65)	1.370	0.938	2.000	0.1035	2.696	1.745	4.166	<0.0001 *
Normal BMI	1	-	-	-	1	-	-	-
Underweight	0.885	0.600	1.305	0.5380	1.081	0.681	1.718	0.7404
Overweight	1.419	0.674	2.986	0.3566	2.138	0.900	5.083	0.0853
Urban	1	-	-	-	1	-	-	-
Rural	1.258	0.873	1.813	0.2173	1.096	0.709	1.695	0.6788
No/occasional smoking	1	-	-	-	1	-	-	-
Regular smoker	1.332	0.847	2.096	0.2148	1.049	0.613	1.798	0.8607
Heavy smoker	1.768	1.099	2.845	0.0188 *	1.253	0.718	2.186	0.4277
No/occasional drinking	1	-	-	-	1	-	-	-
Regular drinker	1.452	0.992	2.127	0.0552	1.404	0.893	2.207	0.1421
Heavy drinker	0.803	0.413	1.560	0.5242	0.966	0.445	2.095	0.9306
General population	1	-	-	-	1	-	-	-
Roma	1.454	0.963	2.195	0.0751	1.366	0.839	2.224	0.2093

*: *p* < 0.05, statistically significant.

**Table 6 viruses-16-00435-t006:** BMI and severity association testing.

	Roma BMI			General Population BMI		
Severity	Underweight	Normal	Overweight	Underweight	Normal	Overweight
Asymptomatic	3 (2.08%)	11 (7.64%)	9 (6.25%)	3 (0.69%)	41 (9.45%)	28 (6.45%)
Mild	3 (2.08%)	17 (11.81%)	11 (7.64%)	7 (1.61%)	59 (13.59%)	48 (11.06%)
Moderate	1 (0.69%)	9 (6.25%)	20 (13.89%)	9 (2.07%)	59 (13.59%)	50 (11.52%)
Severe	1 (0.69%)	22 (15.28%)	31 (21.53%)	10 (2.30%)	65 (14.98%)	49 (11.29%)
Critical	0 (0.00%)	0 (0.00%)	6 (4.17%)	1 (0.23%)	3 (0.69%)	2 (0.46%)
Chi^2^ *p*	0.0347 *			0.9474		
Spearman rho	0.2491			−0.0231		
Spearman *p*	0.0027 *			0.6306		

*: *p* < 0.05, statistically significant.

**Table 7 viruses-16-00435-t007:** Results of the Spearman analysis regarding outcomes and severity.

		rho	95% CI	*p*
Whole lot n = 578	Severity—ICU	0.636	0.584–0.682	<0.0001 *
	Severity—Death	0.435	0.366–0.499	<0.0001 *
General population n = 434	Severity—ICU	0.606	0.543–0.663	<0.0001 *
	Severity—Death	0.412	0.331–0.487	<0.0001 *
Roma n= 144	Severity—ICU	0.700	0.606–0.775	<0.0001 *
	Severity—Death	0.489	0.353–0.604	<0.0001 *

*: *p* < 0.05, statistically significant.

**Table 8 viruses-16-00435-t008:** Correlation analysis between outcomes.

	Roma		General Population	
	ICU-No	ICU-Yes	ICU-No	ICU-Yes
Death-No	84 (58.33%)	26 (18.06%)	284 (65.44%)	70 (16.13%)
Death-Yes	0 (0.00%)	34 (23.61%)	10 (2.30%)	70 (16.13%)
Chi2 *p*	<0.0001 *		<0.0001 *	
Spearman rho	0.658		0.562	
Spearman *p*	<0.0001 *		<0.0001 *	

*: *p* < 0.05, statistically significant.

**Table 9 viruses-16-00435-t009:** Results of logistic regression analysis for elevated inflammatory markers between ethnicities and ICU admission.

Constants (Dependent)	General Population				Roma			
	EXP (β)	95% CI Low	95% CI High	*p*	EXP (β)	95% CI Low	95% CI High	*p*
Elevated CRP	1.110	1.080	1.141	<0.0001 *	1.381	1.194	1.596	0.0003 *
Elevated FER	1.000	0.997	1.003	0.9244	0.999	0.992	1.006	0.7871
Elevated IL-6	1.332	0.997	1.781	0.0523	0.931	0.560	1.550	0.7846
Elevated D-dimers	0.998	0.996	1.001	0.2166	0.998	0.991	1.005	0.5883
Elevated LDH	1.001	0.995	1.007	0.6895	0.992	0.975	1.008	0.3245
Normal HDL	0.947	0.913	0.983	0.0039 *	0.910	0.822	1.008	0.0699
Normal 25-OHD	0.853	0.800	0.910	<0.0001 *	0.864	0.692	1.078	0.1948

CI: confidence interval; β: risk estimate; *: *p* < 0.05, statistically significant.

**Table 10 viruses-16-00435-t010:** Results of logistic regression analysis for elevated inflammatory markers between ethnicities and death.

Constants (Dependent)	General Population				Roma			
	EXP (β)	95% CI Low	95% CI High	*p*	EXP (β)	95% CI Low	95% CI High	*p*
Elevated CRP	1.104	1.076	1.132	<0.0001 *	1.154	1.084	1.229	<0.0001 *
Elevated FER	1.000	0.997	1.003	0.9874	1.004	1.000	1.009	0.0699
Elevated IL-6	1.328	0.946	1.863	0.1007	0.755	0.488	1.167	0.2060
Elevated D-dimers	0.999	0.996	1.001	0.2889	1.005	0.998	1.011	0.1531
Elevated LDH	0.994	0.987	1.001	0.1019	0.991	0.976	1.005	0.2097
Normal HDL	0.920	0.883	0.960	0.0001 *	0.982	0.904	1.066	0.6574
Normal 25-OHD	0.921	0.864	0.983	0.0127 *	0.832	0.703	0.984	0.0318 *

CI: confidence interval; β: risk estimate; *: *p* < 0.05, statistically significant.

**Table 11 viruses-16-00435-t011:** Results of the ROC and AUC analysis in patients admitted to the hospital.

Variable		Roma (n = 144)	Romanian (n = 434)	ICU Related *p* Value	Roma (n = 144)	Romanian (n = 434)	Death Related *p* Value
CRP	AUC	0.985	0.867	<0.0001 *	0.925	0.893	0.2565
	(95% CI)	(0.950–0.998)	(0.832–0.898)		(0.869–0.962)	(0.860–0.920)	
	Cutoff	28.98	29.03		29.4	29.73	
	Sensitivity; Specificity	93.33; 96.43	65.7; 95.9		97.06; 77.27	76.23; 89.55	
FER	AUC	0.796	0.740	0.2196	0.785	0.735	0.3261
	(95% CI)	(0.721–0.858)	(0.696–0.781)		(0.709–0.849)	(0.691–0.776)	
	Cutoff	180.46	201.95		180.46	172.52	
	Sensitivity; Specificity	80.00; 70.24	67.86; 73.13		85.29; 60.00	86.25; 57.63	
IL-6	AUC	0.865	0.860	0.8780	0.764	0.797	0.4738
	(95% CI)	(0.798–0.916)	(0.823–0.891)		(0.687–0.831)	(0.756–0.833)	
	Cutoff	6.46	6.56		6.46	6.66	
	Sensitivity; Specificity	100.00; 64.29	97.86; 62.93		100; 49.11	100.00; 54.00	
D-dimers	AUC	0.781	0.715	0.1534	0.744	0.665	0.1488
	(95% CI)	(0.705–0.846)	(0.670–0.757)		(0.664–0.813)	(0.619–0.710)	
	Cutoff	257.77	286.67		290.55	289.95	
	Sensitivity; Specificity	88.33; 66.67	71.43; 70.41		79.44; 65.53	70.00; 63.28	
LDH	AUC	0.837	0.832	0.8990	0.781	0.763	0.6901
	(95% CI)	(0.766–0.893)	(0.793–0.866)		(0.705–0.846)	(0.720–0.802)	
	Cutoff	196.53	213.69		221.99	225.56	
	Sensitivity; Specificity	85.00; 69.05	87.14; 73.47		85.33; 66.41	78.69; 68.12	
HDL	AUC	0.859	0.797	0.0989	0.772	0.771	0.9893
	(95% CI)	(0.792–0.912)	(0.756–0.834)		(0.695–0.838)	(0.729–0.810)	
	Cutoff	37.83	38.65		34.63	37.83	
	Sensitivity; Specificity	93.33; 70.24	75.71; 72.79		82.35; 68.18	78.67; 68.94	
25-OHD	AUC	0.840	0.845	0.9013	0.788	0.774	0.7684
	(95% CI)	(0.769–0.896)	(0.807–0.877)		(0.712–0.851)	(0.731–0.812)	
	Cutoff	19.93	18.17		19.16	18.82	
	Sensitivity; Specificity	83.33; 78.57	75.00; 80.61		79.41; 68.18	80.00; 58.64	

CRP = C-reactive protein, FER = ferritin, IL-6 = interleukin-6, LDH = lactate dehydrogenase, HDL = high density lipoprotein cholesterol, 25-OHD = 25-hidroxy-vitamin D, *: *p* < 0.05, statistically significant.

## Data Availability

Data are available upon request from the correspondent author.

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
