# Peer review of "Understanding the Impact of COVID-19 on Roma Vulnerable Communities in Western Romania: Insights and Predictive Factors from a Retrospective Study"

_viruses, 2024, doi:10.3390/v16030435_

Round 1

Reviewer 1 Report

Comments and Suggestions for Authors

Congratulations for your work that comes to add certain information for vunerable populations, However I would like to comment a few points that if answered will definitely improve your manuscript.

Comment1: In the abstract body there are noy any summarized conclusions of your study.You only partly refer to methodology followed. I suggest you should clearly state your conclusions because some readers will never go further into your manuscript.

Comment2: In the abstract body should be also mentioned that the results are in comparison to the patients of the general public

Comment3: In the introduction (lines 59-63) you are describing the long covid condition but in your methodology and results nothin similar is reffered.

Comment 4: In the introduction again (lines 76-87) you have a big paragraph in regard to antiinfalmmatory medications tested in severe covid-19, but again in your results nothing analogous is reffered in accordance.

Comment 5: Your study population of 144 Roma patients is very low in order to exclude safe results especially while are further divides in smaller groups for the statistical evaluation. Thetefor tthis has to be stated in the study limitations.

Comment 6: We do know that the obesity status is a very crucial factor for the development of severe Covid19 disease. I suggest then to evaluate if there is a significant difference between the Roma and the general public when statistically test the overweight status with the outcomes.

Comment 7Q According to yout data the Roma population is more overweigt, with increase rate of ICU hospitalization but without significanc in the death rate. This has to be investigated further on in the literature even for other severe contagious diseases and to be included in the discussion section.

Comment 8: Please state any possible study strengths if existed

Comment 9: In Limitations section you are stating again about long covid (lines 378-380) but again nowhere in methodology or results section you are presenting any relevant

Comments on the Quality of English Language

English Language is fine

Author Response

Thank you for the constructive suggestions and ideas. Please find our responses attached.

Reviewer 2 Report

Comments and Suggestions for Authors

The survey of the hospital records for two different population living in the same country is of value considering the isolated Roma social group.

In this paper there are a lot of statistical data (sometimes giving redundant results) but I have some remarks concerning the presentation of the results to be take inot account.

1. Mild diseases are less frequent in the Roma population (evident even if not statistically significant): why? Are patients characteristics identical according to the severity of the disease? The difference of repartition might be related to the willing to attend the hospital which would explain the differences between the two studied populations. If the number of mild and moderate diseases in Roma is underestimated, there is only a marginal difference between the two populations regarding the consequences of the infection.

2. For what reason(s) were asymptomatic patients tested?

3. If all deceased patients were hospitalized in ICU, there is no difference in the death rates of the two populations.

4. The difference in the CRP, IL-6, HDL and 25-OHD values may be merely due to the higher number of severe infection in the Roma population. Is there a correlation between these marker values and the severity of the disease whatever the patient?

The same observation holds for the rural origin and the overweight.

5. While it is said that the Roma population is more susceptible to infectious diseases, are there any data on the prevalence (or incidence) of Covid-19 in this population as compared to the one(s) of other communities?

5. Thus, although the two populations are obviously different considering their social environment, the consequences of the SARS-Coronavirus-2 infections might be the same according to the individual physiological and social status.

Some minor remarks

Ref 1: not accessible; and I am not certain of the affirmation that Covid19 pandemic is of unprecedented scale. The date of the access must be indicated (or another reference has to be included)

Sentences such as “and can be observed in Table 3" should be replaced merely by “(Table 3)” and the text modified accordingly and it is the same for the other tables.

The sentence lines 216-217 “Laboratory ... (calcidiol).” has its place in the material and method section.

Comments on the Quality of English Language

The English must be carefully edited as some mistakes have to be corrected (e.g.: line 164, “satisfactory”, not sufficient; line 167 “data were not”; line 283 “which was done somewhat limited”, of limited access?...)

Author Response

(The authors gave the same response as above.)
